# Extrapolatable Relational Reasoning With Comparators in Low-Dimensional Manifolds

## Abstract

While modern deep neural architectures generalise well when test data is sampled from the same distribution as training data, they fail badly for cases when the test data distribution differs from the training distribution even along a few dimensions. This lack of out-of-distribution generalisation is increasingly manifested when the tasks become more abstract and complex, such as in relational reasoning. In this paper we propose a neuroscience-inspired inductively biased module that can be readily amalgamated with current neural network architectures to improve out-of-distribution (o.o.d) generalisation performance on relational reasoning tasks. This module learns to project high-dimensional object representations to low-dimensional manifolds for more efficient and generalisable relational comparisons. We show that neural nets with this inductive bias achieve considerably better o.o.d generalisation performance for a range of relational reasoning tasks, thus more closely models human ability to generalise even when no previous examples from that domain exist. Finally, we analyse the proposed inductive bias module to understand the importance of lower dimensional projection, and propose an augmentation to the algorithmic alignment theory to better measure algorithmic alignment with generalisation.

## 1 Introduction

The goal of Artificial Intelligence research, first proposed in the 1950s and reiterated many times, is to create machine intelligence comparable to that of a human being. While today's deep learning based systems achieve human-comparable performances in specific tasks such as object classification and natural language understanding, they often fail to generalise in out-of-distribution (**o.o.d**) scenarios, where the test data distribution differs from the training data distribution (Recht et al., 2019; Trask et al., 2018; Barrett et al., 2018; Belinkov & Bisk, 2018). Moreover, it is observed that the generalisation error increases as the tasks become more abstract and require more reasoning than perception. This ranges from small drops (3% to 15%) in classification accuracy on ImageNet (Recht et al., 2019) to accuracy only slightly better than random chance for the Raven Progressive Matrices (RPM) test (a popular Human IQ test), when testing data are sampled completely out of the training distribution (Barrett et al., 2018).

In contrast, human brain is observed to generalise better to unseen inputs (Geirhos et al., 2018), and typically requires only a small number of training samples. For example, a human, when trained to recognise that there is a progression relation of circle sizes in Figure 1a, can easily recognise that the same progression relation exists for larger circles in Figure 1b, even though such size comparison has not been done between larger circles. However, today's state-of-the-art neural networks (Barrett et al., 2018; Wang et al., 2020) are not able to achieve the same. Researchers (Spelke & Kinzler, 2007; Chollet, 2019; Battaglia et al., 2018; Xu et al., 2020) argue that the human brain developed special inductive biases that adapt to the form of information processing needed for humans, thereby improving generalisation. Examples include convolution-like cells in the visual cortex (Hubel & Wiesel, 1959; Güçlü & van Gerven, 2015) for visual information processing, and grid cells (Hafting et al., 2005) for spatial information processing and relational comparison between objects (Battaglia et al., 2018).

In this work, we propose a simple yet effective inductive bias which improves **o.o.d** generalisation for relational reasoning. We specifically focus on a the type of **o.o.d** called 'extrapolation'. For extrapolation tasks, the range of one or more data attributes (e.g., object size) from training and test datasets are

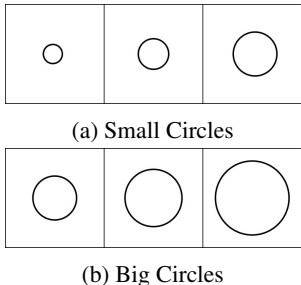

(a) Small Circles

(b) Big Circles

Figure 1: Size Progression Relations for circles of different sizes.

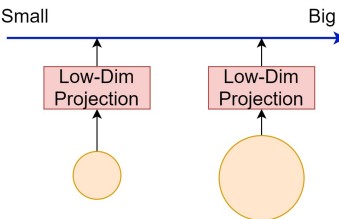

Figure 2: Illustration of projecting object representations onto a 1-dimensional manifold in which size comparison can be achieved by simply measuring the difference between two projections.

completely non-overlapping. The proposed inductive bias is inspired by neuroscience and psychology research (Fitzgerald et al., 2013; Chafee, 2013; Summerfield et al., 2020) showing that in primate brain there are neurons in the Parietal Cortex which only responds to different specific attributes of perceived entities. For examples, certain LIP neurons fire at higher rate for larger objects, while the firing rate of other neurons correlates with the horizontal position of objects in the scene (left vs right) (Gong & Liu, 2019). These observations show that these neurons map objects' attributes such as size and spatial position to the one-dimensional axis of the neurons' firing rate. From a computational perspective, this can be viewed as projecting object representations to different low-dimensional manifolds corresponding to different attributes. Based on these observations (Summerfield et al., 2020), we hypothesise that these neurons are developed to learn low-dimensional representations of relational structure that are optimised for abstraction and generalisation, and the same inductive bias can be readily adapted for artificial neural networks to achieve similar optimisation for abstraction and generalisation.

We test this hypothesis by designing an inductive bias module which projects high-dimensional object representations into low-dimensional manifolds, and make comparisons between different objects in these manifolds. We show that this module can be readily amalgamated with existing architectures to improve extrapolation performance for different relational reasoning tasks. Specifically, we performed experiments on three different extrapolation tasks, including maximum of a set, visual object comparison on dSprites dataset (Higgins et al., 2017) and extrapolation on RPM style tasks (Barrett et al., 2018). We show that models with the proposed low-dimensional comparators perform considerably better than baseline models on all three tasks. In order to understand the effectiveness of comparing in low-dimensional manifolds, we analyse the projection space and corresponding function space of the comparator to show the importance of projection to low-dimensional manifolds in improving generalisation. Finally, we perform the analysis relating to algorithmic alignment theory (Xu et al., 2020), and propose an augmentation to the sample complexity criteria used by this theory to better measure algorithmic alignment with generalisation.

## 2 METHOD

Here, we describe the inductive bias module we developed to test our hypothesis that the same inductive bias of low-dimensional representation observed in Parietal Cortex can be readily adapted for artificial neural network, thus enabling it to achieve similar optimisation for abstraction and generalisation. The proposed module learns to project object representations into low-dimensional manifolds and make comparisons in these manifolds. In Section 2.1 we describe the module in detail. In Section 2.2, 2.3 and 2.4 we discuss how this module can be utilised for three different relational reasoning tasks, which are: finding the maximum of a set, visual object comparisons and Raven Progressive Matrices (RPM) reasoning.

### 2.1 COMPARATOR IN LOW-DIMENSIONAL MANIFOLDS

The inductive bias module is comprised of low-dim projection functions $p$ and comparators $c$. Let $\{o_i; i \in 1 \dots N\}$ be the set of object representations, obtained by extracting features from raw inputs such as applying Convolutional Neural Networks (CNN) on images. Pairwise comparison between object pair $o_i$ and $o_j$ can be achieved with a function $f$ expressed as:

$$f(o_i, o_j) = g( \overset{K}{\underset{k=1}{||}} c_k(p_k(o_i), p_k(o_j))).$$ (1)

Here $p_k$ is the $k^{th}$ projection function that projects object representation $o$ into the $k^{th}$ low dimensional manifold, $c_k$ is the $k^{th}$ comparator function that compares the projected representations, $||$ is the concatenation symbol and $g$ is a function that combines the $K$ comparison results to make a prediction. Having $K$ parallel projection functions $p_k$ and comparators $c_k$ enables a simultaneous comparison between objects with respect to their different attributes. Figure 2 shows an example of comparing sizes of circles by projection onto a 1-dimensional manifold. Both $p$ and $c$ can be implemented as feed forward neural networks. While the comparator $c$, implemented as a neural network, can theoretically learn a rich range of comparison metrics, we found that adding to $c$ an additional inductive bias of distance measure for the projection, such as vector distance $p(o_i) - p(o_j)$ or absolute distance $|p(o_i) - p(o_j)|$, improves the generalisation performance.

Let $r_t(o_i)$ be the ground truth mapping function from $i^{th}$ object's representation $o_i$ to its $t^{th}$ attributes (such as colour and size for a visual object). If such ground truth labels of object attributes exist, $f(o_i, o_j)$ can be trained to directly predict the differences in attributes by minimising the loss $\mathcal{L}(d(r_t(o_i), r_t(o_j)), f(o_i, o_j))$, where $d$ is a distance function (e.g., $r_t(o_i) - r_t(o_j)$ for continuous attributes or $\mathbb{1}_{r_t(o_i)=r_t(o_j)}$ for categorical attributes). However, in real-world datasets, such ground truth attribute labels seldom exist. Instead, in many relational reasoning tasks, learning signals for attribute comparison are only provided implicitly in the training objective. For example, in Visual Question Answering task, an example question might be 'Is the object behind Object A smaller?'. The learning signals for the required size and spatial position comparator are provided only through correctness of the answers to the given questions. Thus, the proposed module is only useful and scalable if it can be integrated into neural architectures for relational reasoning and still learn to compare attributes with the weaker, implicit learning signal. Next, we describe 3 examples of such integrations for different relational reasoning tasks, and show in Section 3 that the proposed module can learn relational reasoning tasks with better generalisation capability.

## 2.2 ARCHITECTURE: MAXIMUM OF A SET

The first task we consider is finding the maximum of a set of real numbers. Formally, given a set $\{x_i; i \in 1 \ldots N\}$ where $x_i$ is a real number represented as a scalar value, we want to train a function $h_{max}(\{x_i, \ldots, x_N\})$ that gives the maximum value in the set. Many neural architectures have been applied on this task, including Deep Sets (Zaheer et al., 2017) and Set Transformer (Lee et al., 2019), but none of them test (**o.o.d**) generalisation capability. In order to test **o.o.d** generalisation in the extrapolation scenario, we create the training and test sets such that their ranges do not overlap. We sample from the range $(V_{low}^{train}, V_{high}^{train})$ for the training set, and from the range $(V_{low}^{test}, V_{high}^{test})$ for the test set, and restrict that $V_{high}^{train} < V_{low}^{test}$.

We integrate the proposed low-dim comparator module with Set Transformer (Lee et al., 2019), a state-of-the-art neural architecture for sets. Set Transformer first encodes each element in the set with respect to all other elements with a Multihead Attention Block (MAB), an attention module modified from self-attention used in language tasks (Vaswani et al., 2017): $e_i = encode(x_i) = \mathsf{MAB}(x_i, x_j)$. The Set Transformer then uses Pooling with Multihead Attention (PMA) to combine all encoded elements of the set as $\mathsf{PMA}(e_1, \ldots, e_N)$. While $\mathsf{MAB}$ uses query and key embeddings to generate attention variables, which are then used as weights in the weight sum of value embeddings of elements, we swap the query-key attention mechanism with our low-dim comparator as:

$$e_i = MLP(\sum_{j=1}^{N} c(x_i, x_j)) \tag{2}$$

Here $c$ is the low-dim comparator and $MLP$ is a standard Multi-Layer Perceptron. Note that we directly use the scalar input $x_i$ as object representation $o_i$ in Equation (1) as no feature extraction is needed. We then use attention-based pooling to combine projection of $x_i$ as $\sum_{i=1}^{N} a(e_i)p(x_i)$, where $a$ outputs attention values while $p$ is the 1-dim projection function. For detailed architecture configuration, please refer to Appendix A.

## 2.3 ARCHITECTURE: VISUAL OBJECT COMPARISON

The second task we consider is comparing visual objects for different attributes such as size and spatial position. For this task, two images $x_1$ and $x_2$ containing single objects of randomly sampled attributes are given, and one is asked if a specific attribute of the second object is larger than, equal to, or smaller than the attribute of the first object. Figure 3a shows an example of this task for comparing

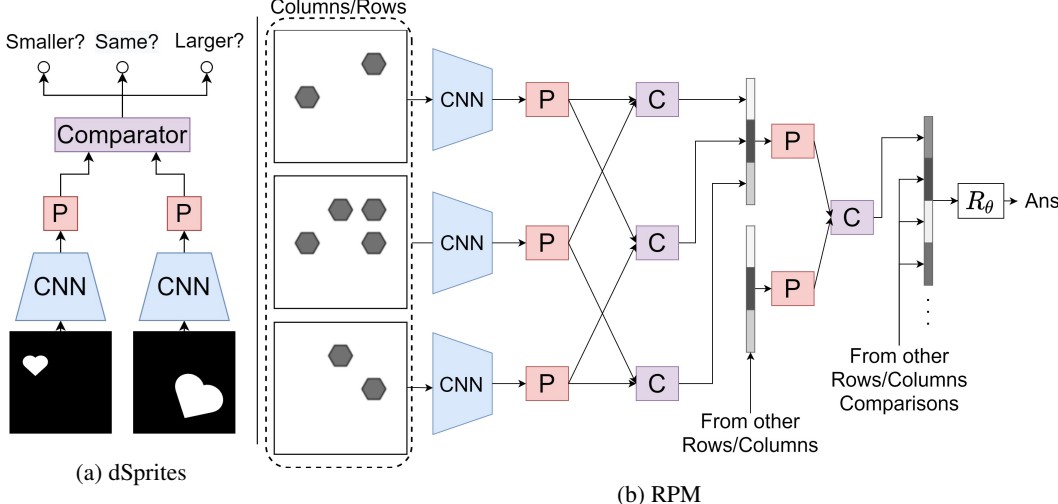

(a) dSprites

(b) RPM

Figure 3: Figure (a) illustrates the architecture for comparing sizes of objects sampled from dSprites dataset. "P" is the projection function. Figure (b) illustrates the architecture for logical reasoning on RPM-style tasks. "P" is the projection function and "C" is the comparator ($g$ in Equation 1).

sizes between two heart-shaped objects. To sample images of objects in the implementation, we used the dSprites dataset (Higgins et al., 2017), a widely used dataset for studying latent space disentangling. To test extrapolation capability, we sample the training set and the test set such that for the compared attribute, the training attribute range has no overlap with the test attributes. We leave details of the dataset construction to Appendix B.

Figure 3a shows an overview of the architecture integrated with the proposed low-dimensional comparator. The image pair $x_1$ and $x_2$ is first passed through a CNN to extract feature embeddings $e_1$ and $e_2$. The feature embeddings are then projected to low-dim manifold and compared as $c(p(e_1), p(e_2))$, where $p$ is the projector and $c$ is the comparator. The comparator has 3 output units with softmax to predict probabilities that an attribute of the second object $a(x_2)$ is smaller than, equal to, or larger than the attribute of the first object $a(x_1)$. The architecture is trained with cross entropy loss with respect to ground truth labels. While we are testing the **o.o.d** generalisation of relational reasoning, it is reasonable to expect that the visual perception module is exposed to all possible scenarios of the input distribution in an unsupervised way. The same assumption also holds for humans, whose vision system has to be exposed to the world inputs sufficiently after birth before they can associate objects with semantic meaning and perform relational reasoning (Maurer, 2016). Thus, we initialise the CNN with the pretrained encoder weights of Beta-VAE (Higgins et al., 2017), a disentangled VAE model trained in the unsupervised setup on dSprites dataset. For detailed configuration of the architecture please refer to Appendix C.

## 2.4 ARCHITECTURE: VISUAL REASONING FOR RAVEN PROGRESSIVE MATRICES

The third task we consider is a more complex visual reasoning task named 'Raven Progressive Matrices' (RPM), which is a popular human IQ test. In this task one is given 8 context diagrams with logic relations present in them, and is asked to pick an answer that best fits with the context diagrams. In our experiment we use the PGM dataset (Barrett et al., 2018), the largest RPM-style dataset available. In the PGM dataset, there is a special data split called 'extrapolation' which is designed to test for the extrapolation scenario of **o.o.d** generalisation.

In this split, colour and size values of objects in the training set are sampled from the lower half of the range, while the same attributes in the test set are sampled from the upper half of the range. Thus the attribute ranges of training and test sets are non-overlapping. For details and examples of the PGM dataset, please refer to Appendix D and Barrett et al. (2018).

Our architecture integrates the low-dim comparator with a Multi-Layer Relation Network (Jahrens & Martinetz, 2020). Figure 3b shows an overview of the architecture developed for PGM tasks. We use a 2-layer relation network with the first layer encoding pairs of diagrams within a row/column and the second layer encoding pairs of rows/columns. Applying such prior knowledge that rules only exist in

rows and columns has been standard practice in state-of-the-art methods for RPM reasoning (Wang et al., 2020; Zhang et al., 2019). Following Wang et al. (2020) we fill each of the 8 candidate answers to the third row and column to obtain in total of 16 answer rows and columns. At each layer of the relation network, we use the low-dimensional comparator instead of the MLP in the original relation network (Santoro et al., 2017) for diagram comparison. Diagram $x_i$ is first passed through a CNN to produce the embedding $o_i$. embedding pairs $(o_i, o_j)$ are then compared as $e_{ij} = f(o_i, o_j)$, where $f$ is the low-dimensional comparator described in Equation (1). The comparison results from the same rows/columns are then concatenated to form row/column embedding $r_{ijk} = e_{ij}||e_{ik}||e_{jk}$. The row/column embeddings are compared with the second layer comparator. The comparison results are then concatenated and input into a reasoning network $R_\theta$ to predict the correct answer. Similar to the dSprites comparison task, as discussed in Section 2.3, we pre-train CNN as encoders of VAE, a technique that has also been previously explored for the PGM dataset (Steenbrugge et al., 2018). For detailed configuration of the architecture, please refer to Appendix E.

## 2.5 Algorithmic Alignment and o.o.d Generalisation

Xu et al. (2020) proposed to measure algorithmic alignment of neural networks to a specific task with sample complexity $\mathcal{C}_\mathcal{A}(g, \epsilon, \delta)$, which is the minimum sample size $M$ so that $g$, the ground truth label mapping function, is $(M, \epsilon, \delta)$-learnable with a learning algorithm $\mathcal{A}$. This essentially says that a model is more algorithmically aligned with a task if it can learn the task more easily with fewer samples. However, in the original definition, both training and test data are independently and identically distributed (i.i.d) samples drawn from the same data distribution. Thus, the algorithmic alignment theory measures how well can a NN fit to a particular data distribution, but does not measure how well can a NN model perform in the **o.o.d** scenario. For example, for the visual object comparison task, an over-parameterised MLP can learn the following two algorithms with low complexity: (1) $m(hash(o_i), hash(o_j))$ where $hash$ is a hashing function and $m$ is a memory read/write function based on the hash index; and (2) $c(p(o_i), p(o_j))$ which is our proposed comparator function. While both algorithms can fit well for the training data, the first algorithm clearly does not **o.o.d** generalise as the memory function does not store unseen samples. In Section 3.5 we confirm experimentally that algorithmic alignment is not indicative of **o.o.d** generalisation.

Intuitively, a more algorithmically aligned model should generalise better as it captures better the underlying algorithm of label generation. Here we propose an augmentation to sample complexity metric (Definition 3.3 in Xu et al. (2020)) in order to measure for algorithmic alignment with generalisation (specifically extrapolation).

**Definition 2.1. o.o.d metric.** Fix an error parameter $\epsilon > 0$ and failure probability $\delta \in (0, 1)$. Suppose $\{x_i^s, y_i^s\}_{i=1}^M$ are i.i.d samples from distribution $\mathcal{D}_s = \mathcal{T}(\mathcal{D}, \beta, \mathbf{u})$, where $\mathcal{D}$ is the full data distribution, $\mathcal{T}$ is a truncating function, $\beta \in (0, 1)$ is the truncation ratio, and $\mathbf{u}$ is the set of dimensions for truncation (See Appendix K for a detailed discussion about truncation function). Let $g$ be the underlying data function $y_i = g(x_i)$, and $f = \mathcal{A}(\{x_i, y_i\}_{i=1}^M)$ be the function learnt with the learning algorithm $\mathcal{A}$. Let $V$ be the value ranges of all dimensions of the data distribution $D$. Then $g$ is $(M, \epsilon, \delta, V, \beta, \mathbf{u}) - learnable$ with $\mathcal{A}$ if:

$$\mathbb{P}_{x \sim \mathcal{D}}[||f(x) - g(x)|| < \epsilon] \geq 1 - \delta \tag{3}$$

The sample complexity is the minimum $M$ for $g$ to be $(M, \epsilon, \delta, V, \beta, \mathbf{u}) - learnable$ with $\mathcal{A}$. In Section 3.5 we show experimentally that this metric measures better a NN's ability to generalise.

## 3 Evaluation

### 3.1 Maximum of a set

For the task of finding the maximum number in a set, we randomly sample number sets of cardinality ranging from 2 to 20 for training, and 2 to 40 for testing. For number sets for training, we uniformly sample numbers in the real value range $[0, 100)$. For testing we sample numbers in the range $[100, 200]$. In this way we test if the model can generalise both, for sets of larger cardinality and for numbers sampled from an unseen range. We sampled 10000 sets for training and 2000 sets for testing. For hyper-parameters of this and subsequent experiments please refer to Appendix F. Table 1 shows the test error of our model compared to Deep Sets (Zaheer et al., 2017) and Set Transformer (Lee et al., 2019), two previous state-of-the-art architectures for sets. Our model achieves a much lower extrapolation error than other methods, even lower than Deep Sets with a built-in Max-Pooling function.

Table 1: Extrapolation test error for learning to find the maximum number in a set of numbers ($mean \pm std$ for 10 runs). M.S.E means Mean Squared Error.

| Model | Deep Sets(Mean) Zaheer et al. (2017) | Deep Sets(Max) Zaheer et al. (2017) | Set Transformer Lee et al. (2019) | OURS |
|-------|---------------------|---------------------|----------------|------|
| M.S.E | $73.22 \pm 17.11$ | $0.51 \pm 0.29$ | $1.62 \pm 0.76$ | $\mathbf{0.0015 \pm 0.0008}$ |

Table 2: Extrapolation test accuracies of baseline and our proposed model for dSprites attribute comparison task (10 runs). X-Coord is horizontal position.

| Attributes / Model | Size | X-Coord | Colour |
|-------|------|---------|--------|
| Baseline | $79.52 \pm 6.71\%$ | $66.14 \pm 5.03\%$ | $78.45 \pm 3.34\%$ |
| OURS | $\mathbf{94.05 \pm 3.03\%}$ | $\mathbf{79.11 \pm 1.92\%}$ | $\mathbf{97.71 \pm 2.81\%}$ |

Table 3: Extrapolation test accuracies for the extrapolation split of PGM dataset.

| Model | WReN Barrett et al. (2018) | MXGNet Wang et al. (2020) | MLRN Jahrens & Martinetz (2020) | MLRN-P | OURS |
|-------|------|--------|------|--------|------|
| Accuracy | 17.2% | 18.9% | 14.9% | 18.1% | **25.9%** |

## 3.2 VISUAL OBJECT COMPARISON

For visual object comparison task, we set three sub-tasks for comparing different attributes of the object, including size, horizontal position and colour intensity. For each task we sample visual objects with different range for the compared attributes from the dSprites dataset (Higgins et al., 2017). Given the compared attribute range $[V_{low}, V_{high}]$, we sample training data from range $[V_{low}, \frac{2}{3}V_{high})$ and test data from range $[\frac{2}{3}V_{high}, V_{high}]$. As ground truth attribute value is provided in the dSprites dataset, we can build the comparison label as $(\mathbb{1}_{a_1<a_2}, \mathbb{1}_{a_1=a_2}, \mathbb{1}_{a_1>a_2})$. For all experiments we sample 60000 training pairs and 20000 test pairs. We test our proposed model against an MLP baseline, which directly processes the object representations $o_i$ and $o_j$ extracted by CNN as $MLP(o_i, o_j)$. We select the best MLP by hyper-parameter search over the number of layers and layer sizes. We use a 1-dimensional projection as we find this gives the highest accuracy. Table 2 shows the extrapolation test accuracies of our model compared to the baseline. Our model with low-dimensional comparator significantly outperforms baselines for all three compared attributes.

## 3.3 VISUAL REASONING FOR RAVEN PROGRESSIVE MATRICES

For RPM-style tasks we use the extrapolation split of the PGM dataset (Barrett et al., 2018), which is already a well-defined extrapolation type of generalisation task. We compare our proposed model against all previous methods (to the best of our knowledge) that have reported results on the extrapolation data split. We additionally include a baseline model named "MLRN-P", which is a 2-layer MLRN (Jahrens & Martinetz, 2020) with prior knowledge of only the relations present in rows/columns and with pre-training. Table 3 shows the test accuracy comparison. Our proposed model outperforms all other baselines. We note that we used a vanilla CNN as the perception module, same as most previous methods (Barrett et al., 2018; Zhang et al., 2019; Jahrens & Martinetz, 2020) on RPM tasks. Multiple-object representation learning methods (Greff et al., 2019; Kosiorek et al., 2019; Wang et al., 2019a), which achieve better results for multi-object scene learning than CNN, can be investigated for potential improvement of generalisation performance. We leave this for future work.

## 3.4 WHY LOW DIMENSION?

While we show that the comparators in low-dimensional manifolds improve **o.o.d** generalisation for a range of relational reasoning tasks, the reason behind it is still not clear. In this section we analyse the projection space and the comparator function landscape of the visual object comparison task to shed light on this. We first state 3 observations invariant across different sub-tasks comparing different attributes:

1. *When the ground truth attribute can be represented in a 1-dimensional manifold (such as vertical position), comparators in higher dimensions learn to project the object representation into the 1-dimensional manifold.* Figure 4a illustrates this with a plot of the projection

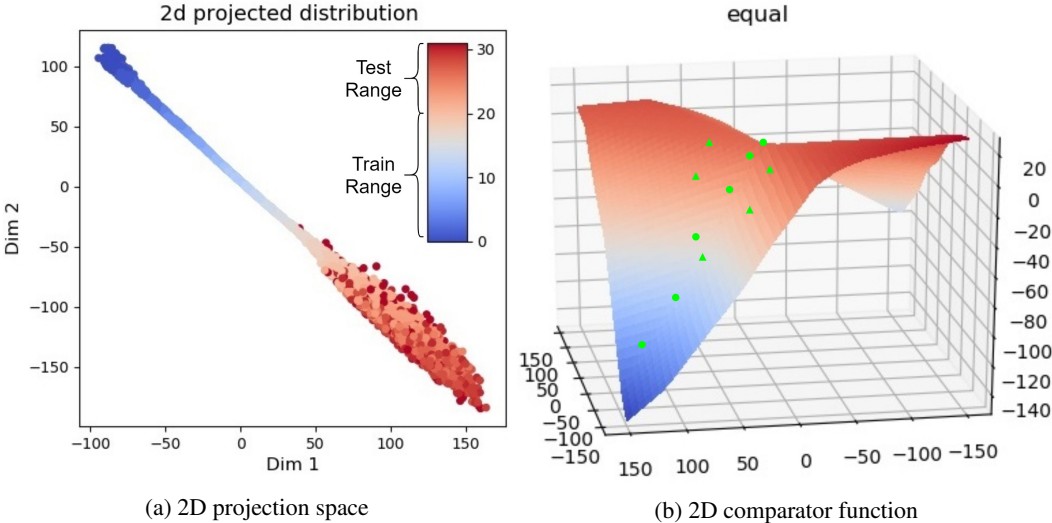

(a) 2D projection space

(b) 2D comparator function

Figure 4: (a) shows a scatter plot of the 2D projected distribution of objects in the task of comparing x-coord attributes of objects in the images. X-axis and Y-axis are 2 dimensions of the projection manifold. Latent vertical position (ranging from 0 to 32) is indicated by colour. (b) plots the comparator's function landscape for "equal" output unit (before softmax) in the space of vector differences between 2D projections of objects. Green circles represent the vector difference sampled from the training set while triangles represent the vector difference sampled from the test set.

> distribution for the task of comparing vertical positions. It can be observed that even though the projection space is 2-dimensional, the projected points cluster around a line.

2. *The projection of the test data in the manifold is less clustered around the sub-manifold of attributes than that of training data.* This can be observed from Figure 4a, where the points projected from the test set are more spread out than from the training set. There is also less order in the distribution of test points, where points of noticeably different intensity appear next to each other.

3. *The function landscape becomes less defined outside of the sub-manifold.* Figure 4b plots the comparator function landscape for the "equal" ($\mathbb{1}_{a_1=a_2}$) output unit over the space of vector differences between 2D projected representation $p(o_2) - p(o_1)$. The equality function is well defined in the sub-manifold in which training points (circles) lie, peaking close to the $(0, 0)$ point. However, outside of the training points' sub-manifold, the function is more random, with a significant region with higher function value than at $(0, 0)$ point. Note that the vector differences of test points (triangles) may be in this region.

From the above observations, we conclude that when comparators are of higher dimension than the intrinsic dimension of compared attributes, the projection tends to lie in a sub-manifold of the same dimension as for the attributes, resulting in the comparator function to be only well defined in that manifold. However, the projection of test data tends to escape from this sub-manifold into the region where the comparator function is never trained on, resulting in incorrect prediction.

In addition to the observations above, we also use Hausdorff distance as a quantitative measure of discrepancies between projected repesentations from the training and test sets. Due to space limitations, this is discussed in Appendix H.

We also performed ablation study on the dimensionality of the projected embedding space. Figure 5 shows the plot of test accuracy against different dimension size of the projection functions for the visual object comparison tasks. It can be observed that increasing dimension sizes (x-axis) reduce the test performance. This further validate that low-dimensional embeddings are crucial in improving neural architecture's extrapolation performance.

## 3.5 ALGORITHMIC ALIGNMENT

Figure 6a shows the **o.o.d** and **i.d** (identically distributed) test accuracies of baseline and our proposed comparator model for size comparison task with different training sample sizes. The dataset, model

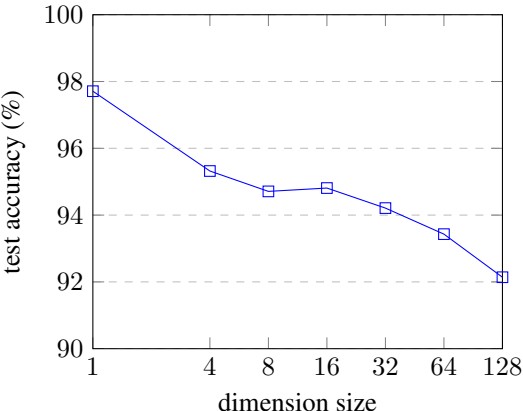

Figure 5: Projection dimension ablation study for visual object comparison tasks.

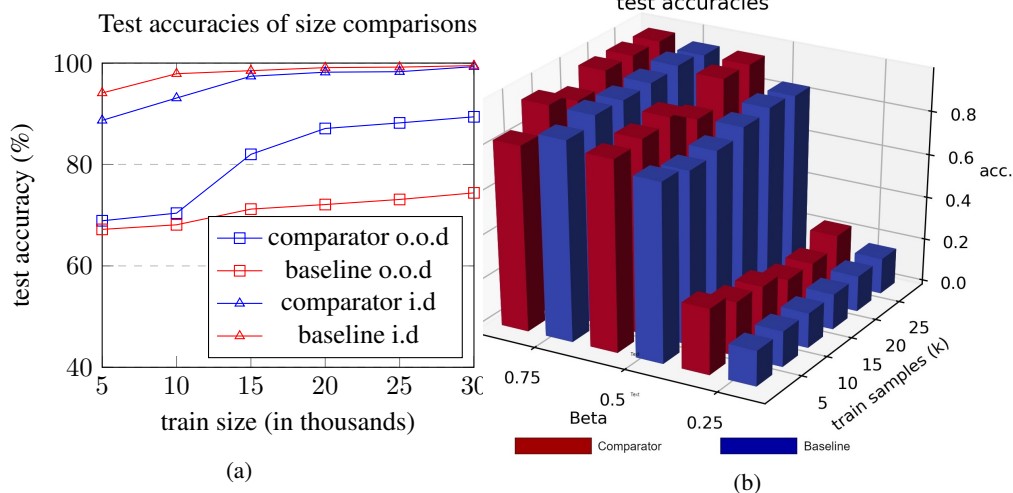

Figure 6: Figure (a) shows o.o.d and i.d. (identically distributed) test accuracy of baseline and comparator for different training sample sizes. (b) shows test accuracies of baseline and comparator for differnt training sample sizes and with different $\beta$-rate for truncating training distribution.

and baseline configurations are the same as in Visual Object Comparison task (Section 2.3). It can be observed that **i.d** test accuracy's sample complexity (training samples needed to achieve the same accuracy), which measures algorithmic alignment, is not indicative of **o.o.d** test accuracies.

Figure 6b shows the size comparison of the test accuracies of baseline and comparator for different training sample sizes with different $\beta$ rates for truncating the training distribution along the latent dimension 'size'. For example when $\beta = 0.5$, only objects with size in the range $[V_{low}, 0.5V_{high}]$ is seen during training. This corresponds to our proposed metric in Section 2.5. The new metric reflects that the model which learns better with truncated training distribution is the one with better **o.o.d** generalisation.

## 4 RELATED WORK AND CONCLUSION

**o.o.d Generalisation**: The deep neural network's lack of **o.o.d** generalisation (sometimes termed domain generalisation or extrapolation) capability has recently come under scrutiny. Different types of approaches have been proposed to improve **o.o.d** generalisation, such as reducing superficial domain specific statistics of training data (Wang et al., 2019b; Carlucci et al., 2019), adversarially learn representations that are domain-invariant (Li et al., 2017; Albuquerque et al., 2019), disentangling representations to separate functional variables with spurious correlations (Heinze-Deml & Meinshausen, 2017; Gowal et al., 2019), and constructing models with innate causal inference graphs to reduce the dependence on spurious correlations (Arjovsky et al., 2019; Bengio et al., 2020). Our work aligns more with the line of work on discovering inductive bias that improves generalisation.

Arguably CNN (LeCun et al., 1995) is such an inductive bias that improves generalisation on visual input, and Graph Neural Network an inductive bias on graph-structured data (Battaglia et al., 2018). Trask et al. (2018) proposed Neural Arithmetic Logic Units (NALU), an inductive bias that allows neural networks to learn simple arithmetic with improved **o.o.d** generalisation.

**Relational Reasoning**: There is a wide range of relational reasoning tasks such as Visual Question Answering (Antol et al., 2015; Johnson et al., 2017), Raven Progressive Matrices (Barrett et al., 2018; Zhang et al., 2019), and Inferring Physical interactions (Kipf et al., 2018; Sanchez-Gonzalez et al., 2018). These tasks involve comparing entities such as visual objects to infer relations between them. A large proportion of models proposed to solve relational reasoning tasks fall into the broader range of graph neural networks and relational networks (Battaglia et al., 2018). Our work is mostly orthogonal to these works, and may be viewed as a special type of edge layer that can be integrated into most of these models. There is another line of research on neural-symbolic models (Yi et al., 2018; Mao et al., 2019). Our work differs from these approaches in not using pre-defined programs, but learnable modules for comparing representations. PrediNet (Shanahan et al., 2019) use self-attention to extract features from the input, and then compare extracted features with a comparator. While our work is similar to PrediNet in that we also use a comparator module to compare extracted features, PrediNet uses a different comparator module, different ways to extract features, and does not explicitly investigate how low-dimensional projections help with **o.o.d** generalisation.

**Low-Dimensional Embedding**: While there is a rich history of producing low-dimensional embeddings for machine learning (see Appendix J for a brief discussion), to the best of our knowledge, little prior work has investigated how low-dimensional embedding improves **o.o.d** generalisation for relational reasoning tasks. van Steenkiste et al. (2019) is the most related work, but focuses on the disentanglement property rather than on low-dimensional property for relational reasoning tasks.

**Conclusion**: We proposed a neuroscience-inspired inductive bias for improving the generalisation ability of neural networks for relational reasoning tasks. We showcased its effectiveness on three selected tasks, but the method can readily be adapted to any other relational reasoning task.

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

## A  MAXIMUM OF A SET: ARCHITECTURE CONFIGURATIONS

Table 4: Model configuration for Maximum of a Set task.

| $K$ | Projection Dim | Comparator Type | Number of Objects | Output Type | Loss Function |
|---|---|---|---|---|---|
| 1 | 1 | $MLP(p(o_i) - p(o_j))$ | 20 | Real-Valued | M.S.E |

The architecture for the maximum of a set task has three sub-modules, namely a comparator $f(x_i, x_j)$, a comparison summariser $e_i = MLP(\sum_{j=1}^{N} f(x_i, x_j))$ and a pooling function $\sum_{i=1}^{N} a(e_i)p(x_i)$. For comparator $f$, we set $K$, the number of parallel comparisons (Equantion 1 in the main body of the paper) to be 1 because the scalar valued real numbers do not have multiple parallel attributes. We implement the projection function $p$ as a single feed forward layer. We choose 1-dimensional comparison space as this gives the best result. The comparison function $c$ takes the projected difference $p(x_i) - p(x_j)$ as input, and is implemented as a single feed forward layer with 1 output unit. The $MLP$ in the comparison summariser is implemented as a 2-layer MLP of hidden-size $16 - 1$. In the pooling function, the attention function $a$ is implemented as a softmax layer, which normalises $e_i$ across $i \in 1 \ldots N$. Table 4 shows summarised model configuration.

## B  VISUAL OBJECT COMPARISON: DATASET GENERATION

In this section we describe details of the visual object comparison dataset. We sample images from the dSprites dataset Higgins et al. (2017) and generate comparison labels (categories include smaller than, equal to, greater than) from ground truth latent values provided in the dataset. For each image in the dSprites dataset, 5 ground truth attribute values are provided, which are shape "category", "size", "rotation angle", "horizontal position" and "vertical position". We add "colour" as the 6th attribute by randomly generating the colour intensity value from the set [0.1,0.2,0.3,0.4,0.5,0.6,0.7,0.8,0.9,1.0] for each image. We multiply image pixel values with the colour intensity, and add the colour intensity value to the ground truth latent values.

Algorithm 1 shows the pseudo-code for generating the visual object comparison dataset. compare_attr indicates the object attribute to be compare for the task. We pick three different attributes for experiments, which are "size", "horizontal position" and "colour intensity". We set training attribute range to be the lower 60% while test attribute range to be the upper 40%.

**Input:** train_size, test_size, compare_attr
train_data = EmptyList()
test_data = EmptyList()
**for** split in {train,test} **do**
  **for** i = 1 to train_size **do**
    attr_range, attr_idx = attr_stats(compare_attr)
    **if** split == train **then**
      LO, HI = 0, 0.6
    **else**
      LO, HI = 0.6, 1.0
    **end if**
    sample_range = Truncate(attr_range, LO, HI)
    latent_values_A = SampleLatent(attr_idx, sample_range)
    latent_values_B = SampleLatent(attr_idx, sample_range)
    image_A = SampleImage(latent_values_A)
    image_B = SampleImage(latent_values_B)
    less_than = latent_values_A < latent_values_B
    equal = latent_values_A == latent_values_B
    greater_than = latent_values_A > latent_values_B
    label = Concat(less_than, equal, greater_than)
    **if** split == train **then**
      train_data ← (image_A,image_B,label)
    **else**
      test_data ← (image_A,image_B,label)
    **end if**
  **end for**
**end for**

**Algorithm 1:** Visual Object Comparison Dataset Generation

## C VISUAL OBJECT COMPARISON: ARCHITECTURE CONFIGURATIONS

In this section we list detailed configuration of the architecture with low-dim comparator and the baseline architecture. The architecture with low-dim comparator is illustrated in Figure 3a. The CNN module is a 4-layer CNN of filter number $32 - 32 - 64 - 64$ followed by a 2-layer MLP of hidden size $256 - 10$. Each CNN layer has stride value 2 and padding value 1. The output of each CNN is the object representation $o_i$ of raw image input $x_i$. The projection module $p$ is a single feed forward layer projecting to a space with dimension $d$. We use $d = 1$ for reporting results except for the manifold analysis experiments in Section 3.4. The comparator takes projected vector difference $p(o_i) - p(o_j)$ as input, and is implemented as a 2-layer MLP of size $h - 3$, where $h$ is the hidden size and 3 is the output size (corresponding to 3 different output categories). We found that varying $h$ in the range 4 to 32 has little effect on the performance. Hence we report performance with the $h = 4$ to reduce computational costs.

The baseline architecture uses the same CNN module as the architecture with low-dim comparator. The baseline model concatenates the CNN output $o_i$ and $o_j$ and feed the concatenated vector into a MLP. We performed hyper-parameter search of the MLP with number of layers ranging from 1 to 4, and with hidden unit sizes from 32 to 64. We found that $64 - 64 - 3$ is the best performing architecture.

Table 5: Model configuration for Visual Object Comparison task.

| $K$ | Projection Dim | Comparator Type | Number of Objects | Output Type | Loss Function |
|---|---|---|---|---|---|
| 1 | 1 | $MLP(p(o_i) - p(o_j))$ | 2 | Categorical | Cross-Entropy |

## D PGM DATASET

In this section we give a brief description of PGM dataset. For more details please refer to Barrett et al. (2018). PGM contains 8 context panels and 8 answer panels. The 8 context panels for a $3 \times 3$ diagram matrix. One is asked to pick the answer the logically fit with the context panels. In PGM, logic relations can exist in both rows and columns of the diagram matrix. Figure 7a and 7b show two examples from the PGM dataset(Image courtesy Barrett et al. (2018)). The first example contains a 'Progression' relation of the number of objects across diagrams in columns. The second examples contains a 'XOR' relation of position of objects across diagrams in rows. The objects in PGM datasets have different attributes such as colour and size. In total five types of relations can be present in the task: $\{Progression, AND, OR, XOR, ConsistentUnion\}$.

## E PGM ARCHITECTURE CONFIGURATIONS

In this section we describe detailed configuration of the PGM architecture with low-dim comparators, and the baseline model MLRN-P, which is an augmented version of MLRN Jahrens & Martinetz (2020). The descriptions here is supplementary to descriptions in section 2.4 of the main paper and figure 3b.

The CNN module is a 4-layer CNN of filter number $32 - 32 - 64 - 64$ followed by a 2-layer MLP of hidden size $256 - 128$. Each CNN layer has stride value 2 and padding value 1. The output of each CNN is the object representation $o_i$ of raw image input $x_i$. Following Barrett et al. (2018) we attach to $o_i$ a position tag to indicate its position in the diagram matrix. The tagged object representation is then projected onto $K = 512$ 1-dimensional manifold for parallel attribute comparison. Next we describe the comparator module $f$ (equation 1. As shown in figure 3b, there two hierarchical projection comparison. For the first level, we found that implementing $c$ as a simple vector difference module achieve best results, which means $c = p(o_i) - p(o_j)$. This reflects the fact that comparison between diagrams is directional, such as increase in number of objects from one diagram to the other. We implement $g$ in equation 1 as a residual MLP of 4 layers with hidden size $2048 - 2048 - 2048 - 796$. The output from each pairwise comparison of diagrams in a row/column are concatenated to form the relation embedding for that row/column. For the second level we implement $c$ as absolute difference, which means $c = |p(o_i) - p(o_j)|$. This works better because relation comparison is less directional. For example the difference between relation of increasing number of objects and relation of increasing sizes of objects should be the same when the compared diagram is swapped. For the second level we implement $g$ as 3 layer MLP of hidden sizes $1024 - 512 - 1$, which directly output the predicted similarity score between two rows/columns. For predicting the correct answer candidate we follow Barrett et al. (2018) by applying a softmax function to scores produced by comparing each

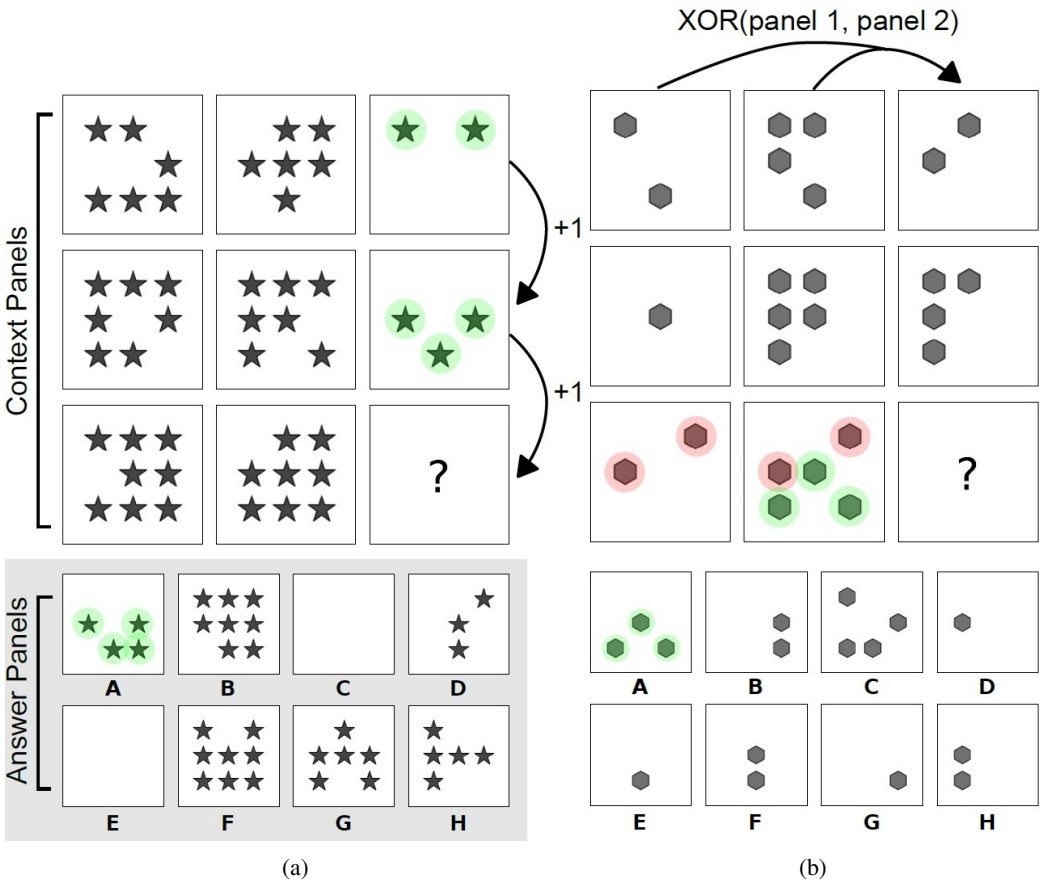

Figure 7: Two examples in PGM dataset. (a) task contains a 'Progression' relation of the number of objects across diagrams in columns while (b) contains a 'XOR' relation of position of objects across diagrams in rows. For both task "A" is the correct answer.

answer rows/columns with context row/columns to produce scores for each answer candidates. For meta-target prediction we sum all context row/column embedding and process it with a 3-layer MLP of hidden size $1024 - 512 - 12$, where 12 is the meta-target label size. We have attached source code in the submission for clarity.

The baseline model MLRN-P is modified from MLRN Jahrens & Martinetz (2020). We have performed three architecture modifications for fair comparison with our model. Firstly we inject the prior knowledge of relations existing only in rows/columns into the model. The first level of Relation Networks compare diagrams within row/columns and the second level of Relation Networks compare row/colum embeddings. Secondly we swap MLP in the original MLRN with residual MLP, which is shown to improve performance slightly in our model. Thirdly we pretrained CNN module with Beta-VAE Higgins et al. (2017) for fair comparison with our model.

## F    TRAINING DETAILS

In this section we describe the training details for all three experiments. We use PyTorch[1] for implementation. For gradient descent optimiser, we use RAdam Liu et al. (2019), an improved version of the Adam optimiser. For all 3 experiments we use learning rate 0.001 and betas (0.9,0.999). We used 2 Nvidia Geforce Titan Xp GPUs for training all models. For Maximum of a set and visual object comparison, we set batch size to be 64. For PGM we found a larger batch size of 512 slightly improves result. For Maximum of the set and visual object comparison we set training epochs to be 20. For PGM we trained for 50 epochs. For visual object comparison and RPM tasks we pre-trained

---

[1]https://pytorch.org/

CNN as the encoder of a Beta-VAE Higgins et al. (2017). We follow standard training procedures of Beta-VAE as described in the paper, and set the $\beta$ value to be 1.

Table 6: Model configuration for Maximum of a Set task.

| $K$ | Projection Dim | Comparator Type | Number of Objects | Output Type | Loss Function |
|-----|----------------|-----------------|-------------------|-------------|---------------|
| 512 | 1 | $MLP(p(o_i) - p(o_j))$ | 16 | Categorical | Cross-Entropy |

## G  ADDITIONAL PLOTS

In section 3.4 of the main paper we show plots of projected distribution and comparator's function landscape for position comparison task. In this section we show the same plots for comparison tasks for other attributes, which are sizes and colour intensity. Figure 8 shows the plot for size comparison while figure 9 shows the plot for colour intensity comparison. Training range is the lower 60% of the colour bar while test range is the upper 40%. The observations stated in section 3.4 also holds true for these attributes. For size and colour intensity comparison tasks, the projected distribution plots show that the test data is more clustered than that of spatial position comparison. This shows that size and colour intensity attributes can be learnt better with a bespoke CNN perception module than spatial position attributes. This is supported by that models trained for these two attributes achieved higher **o.o.d** test accuracies.

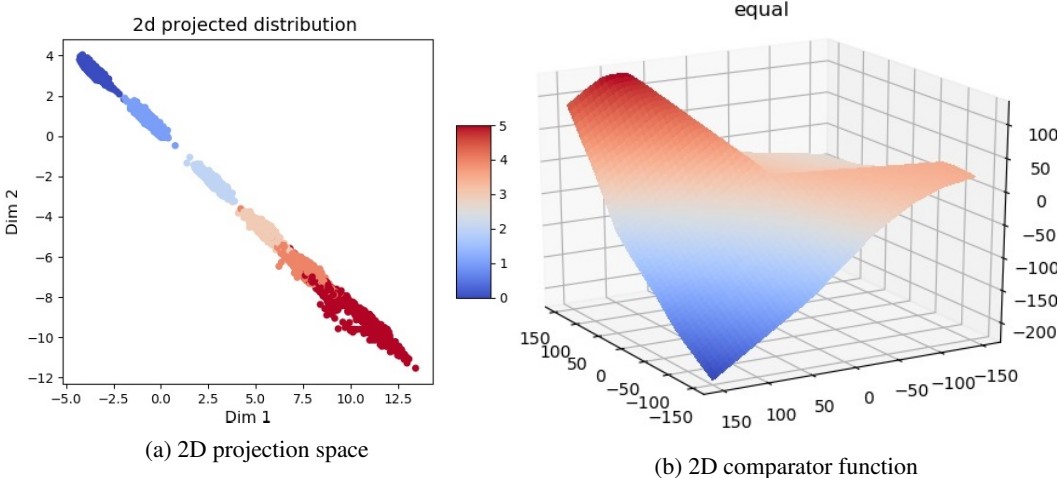

(a) 2D projection space

(b) 2D comparator function

Figure 8: (a) shows a scatter plot of the 2D projected distribution of objects in the task of comparing sizes of objects in the image. X-axis and Y-axis are 2 dimensions of the projection manifold. Latent size variable (possible values are [0,1,2,3,4,5]) is indicated by colour. (b) plots the comparator's function landscape for "equal" output unit (before softmax) in the space of vector differences between 2D projections of objects.

## H  QUANTITATIVE EVIDENCE FOR LOW DIMENSION

In section 3.4 and G we showed visual illustration for 1-dim and 2-dim comparator's function landscape for the visual object comparison task to illustrate how out-of-distribution test data lies outside of the training function landscape. It is difficult to visualise function landscape of higher dimensions. In this section we discuss how to use Hausdorff Distance, which measures distance between sets, to measure the differences between projected distance from training and **o.o.d** test set.

Hausdorff distance measures how far subsets of a metric space are from each other. Given two sets $A = \{a_1, a_2, \ldots, a_N\}$ and $B = \{b_1, b_2, \ldots, b_N\}$ of the same metric space. The Hausdorff distance of $A$ and $B$ is computed as:

$$d_H(A, B) = \max \left\{ \sup_{a \in A} \inf_{b \in B} \mathrm{d}(a, b), \sup_{b \in B} \inf_{a \in A} \mathrm{d}(a, b) \right\} \tag{4}$$

where $d$ is a distance measure of the metric space. Informally Hausdorff distance measures the greatest of all the distances from a point in one set to the closest point in the other set.

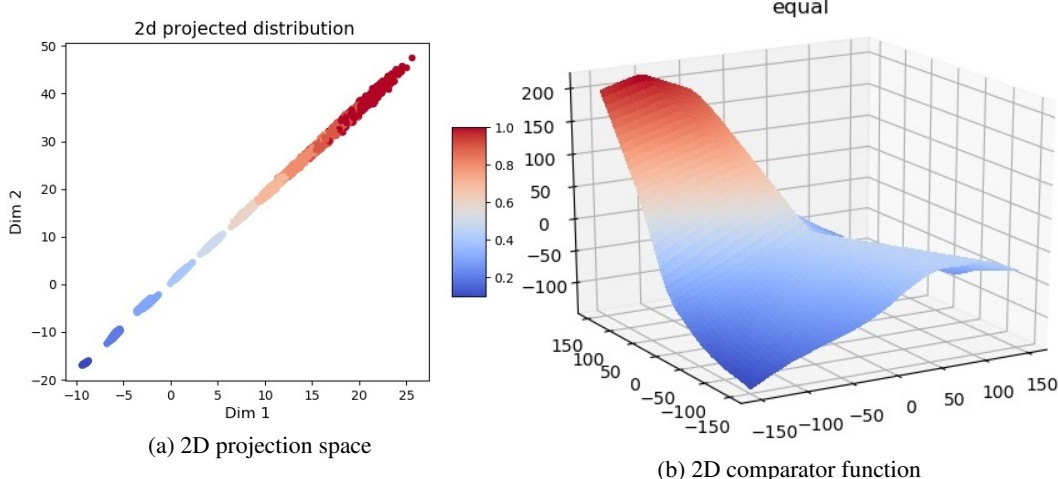

(a) 2D projection space

(b) 2D comparator function

Figure 9: (a) shows a scatter plot of the 2D projected distribution of objects in the task of comparing colour intensity of objects in the image. X-axis and Y-axis are 2 dimensions of the projection manifold. Latent colour intensity variable (possible values are [0.1,0.2,0.3,0.4,0.5,0.6,0.7,0.8,0.9,1.0]) is indicated by colour. (b) plots the comparator's function landscape for "equal" output unit (before softmax) in the space of vector differences between 2D projections of objects.

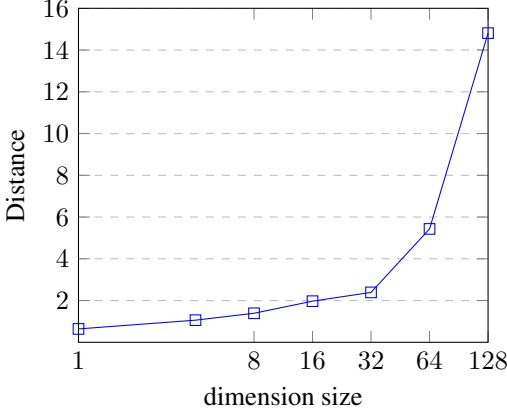

Figure 10: Hausdorff distance between training set and test set of vector differences between projected object representations (normalised) for the visual object comparison task.

We apply Hausdorff distance to measure distances between sets of vector differences from training and test set. The training set $S_{train}$ contains vector difference $p(e_i) - p(e_j)$ for all pairs of inputs $e_i, e_j \in \mathcal{D}_{train}$ from the training distribution $\mathcal{D}_{train}$. Recall that $p$ is the projection module. The test set $S_{test}$ contains vector differences of pairs from the test data distribution $\mathcal{D}_{test}$. We measure the Hausdorff distance between training and test set $d_H(S_{train}, S_{test})$ for different projection dimensions, and show the result in Figure 10. To compare in the same scale, we normalise vector differences with the mean and standard deviation of the training set. It can be observed that as the dimensionality of the projection space increase, there is increased distance between $S_{train}$ and $S_{test}$. This quantitatively shows that the observations in Section 3.4 extends beyond 2-dimensional cases.

## I  ABLATION STUDIES

We performed ablation studies of different hyper-parameters in our experiments. and show the results in Table 7, Table 8 and Table 9. Table 7 shows the test accuracy of architectures with different number of projection functions for PGM task. The number of projection functions is indeed another hyper-parameter to tune, but we think the improved performance is definitely worth it. Table 8 shows the test accuracy on the extrapolation split of PGM dataset with different pre-training for the perception module. We picked the hyper-parameters giving the best extrapolation test accuracy.

Table 9 shows the effectiveness of different comparator input types on the test accuracy for comparing 'colour' attributes in the Visual Object Comparison task.

| Number of Projector Functions | Accuracy |
|:---:|:---:|
| 128 | 20.1 |
| 256 | 24.2 |
| **512** | 25.9 |

Table 7: Ablation Study on Number of Projector/Comparator functions for PGM task.

| Pre-trained Encoder | Accuracy |
|:---:|:---:|
| $\beta$-VAE ($\beta$=4, dim=64) Steenbrugge et al. (2018) | 23.5 |
| $\beta$-VAE ($\beta$=4, dim=128) | 25.1 |
| $\beta$-VAE ($\beta$=1, dim=128) (OURS) | 25.9 |

Table 8: Ablation study on different pre-trained encoders for PGM task.

vector dist accuracy:0.9771 +- 0.0281 l1 dist accuracy:0.692std +- 0.149 MLP dist accuracy:0.912 +-0.0652

| Comparator Input Type | Accuracy |
|:---:|:---:|
| Vector Difference | $97,71 \pm 2.81\%$ |
| L1 Difference | $69.2 \pm 14.9\%$ |
| Concatenation | $91.2 \pm 6.52\%$ |

Table 9: Ablation study on different input types of comparator functions for 'colour' comparison task.

## J   A BRIEF DISCUSSION ON LOW-DIMENSIONAL EMBEDDING

In machine learning there is a rich history of learning low-dimensional embedding, ranging from more classic methods like Principal Component Analysis (PCA) (Wold et al., 1987) and Independent Component Analysis (ICA) (Comon, 1994) to recent neural-network-based methods like Variational Auto-Encoders (VAE) (Kingma & Welling, 2013) and Generative Adversarial Networks (GAN) (Goodfellow et al., 2014). VAE and GAN have been frequently used to learn in an unsupervised manner the low-dimensional representations from an unlabelled dataset, and use the learnt representations to boost performance of supervised downstream tasks, such as image classification and natural language processing. However, most researches on VAE and GANs put more emphasis on understanding how disentanglement affect downstream task performance, but rather than low-dimensions, such as Beta-VAE (Higgins et al., 2017) and its variants. Arguably disentanglement can be considered as splitting the latent distribution into multiple orthogonal distributions of lower dimensions, but to our best knowledge this view has not been discussed in the literature. Moreover, there is limited work on how low-dimension representations learnt by neural generative models help with **o.o.d** generalisation. To our best knowledge, the only work discussing this is by Zhao et al. (2018), who systematically study the generalisation capability of GANs and VAEs using experimental methods from cognitive psychology.

## K   TRUNCATION FUNCTION

The truncation function $\mathcal{T}(\mathcal{D}, \beta, \mathbf{u})$ in section 2.5 can be any functions that truncate the data distribution $\mathcal{D}$ given a truncation ratio $\beta$ and truncation dimensions $\mathbf{u}$. The truncation can be performend in the data distribution (e.g. select one dimension of the data space and truncate it to be in range $(0, \beta)$. In the 'Visual Object Comparison' task, we truncate the latent data distribution instead. The latent data distribution of dSprites dataset(according to the beta-VAE and dSprites dataset paper) has six latent dimensions which are: (color, shape, size, rotation, x-coord, y-coord). For example, we can select the x-coord dimension (range [1,32]), and truncate the distribution with $\beta = 0.6$ so the samples from the truncated distrubtion can only have x-coord in range[1,32*0.6].

## L   CORRELATION BETWEEN PROJECTION AND GROUND TRUTH ATTRIBUTES

We analysed the projected 1-D embeddings by our model for the Visual Object Comparison task to see if they correlate with the ground truth attributes. We measured Pearson Correlation between

projection and ground truth attribute values for all three comparison tasks. The results are listed in Table 10. It can seen that our model learns projections that are highly correlated with the ground truth attribute values, even though the ground truth is never used during training.

| Attribute Type | Correlation Score |
|:---:|:---:|
| Size | 0.945 |
| Colour | 0.977 |
| X-Coord | 0.796 |

Table 10: Pearson Correlation between projected embedding and ground truth attribute values.

## M    RESULTS ON OTHER PGM SPLITS

We additionally tested our model on "neutral" and "interpolation" split of PGM dataset, and compare our model's results with other baselines' in Table 11. Our model performs better than other baselines for "interpolation" task, while is outperformed by MLRN Jahrens & Martinetz (2020) on "neutral" split.

Table 11: Test accuracies for "neutral" and "interpolation" split of PGM dataset.

| Data Split | WReN Barrett et al. (2018) | MXGNet Wang et al. (2020) | MLRN Jahrens & Martinetz (2020) | OURS |
|:---:|:---:|:---:|:---:|:---:|
| Neural | 76.9% | 89.6% | **98.0%** | 90.1% |
| Interpolation | 67.4% | 84.6% | 57.8% | **85.4%** |

## N    MAXIMUM OF A SET: ADDITIONAL BASELINE

For the "Maximum of a set" experiment, we additionally tested a Relation-Network based model that explicitly takes the comparison results as input. The Relation Network first encoded pairwise between all pairs of element $(x_i, x_j)$ using a MLP (32-32) taking the concatenation of element values as input. For each element $x_i$ we sum all pairwise embeddings between this element and others elements $x_j$ as $e_i = \sum_j MLP(x_i, x_j)$. The embeddings $e_i$ are then passed through a max-pooling layer, and then another MLP to produce the final output. This model achieves $16.2 \pm 2.45$ M.S.E, which is very large and thus are not included in the main paper.

## O    ADDITIONAL PLOTS OF COMPARATOR FUNCTION LANDSCAPE

Figure 11 shows plots of the 'equal' comparator function landscape for independent runs of x-coord comparison task.

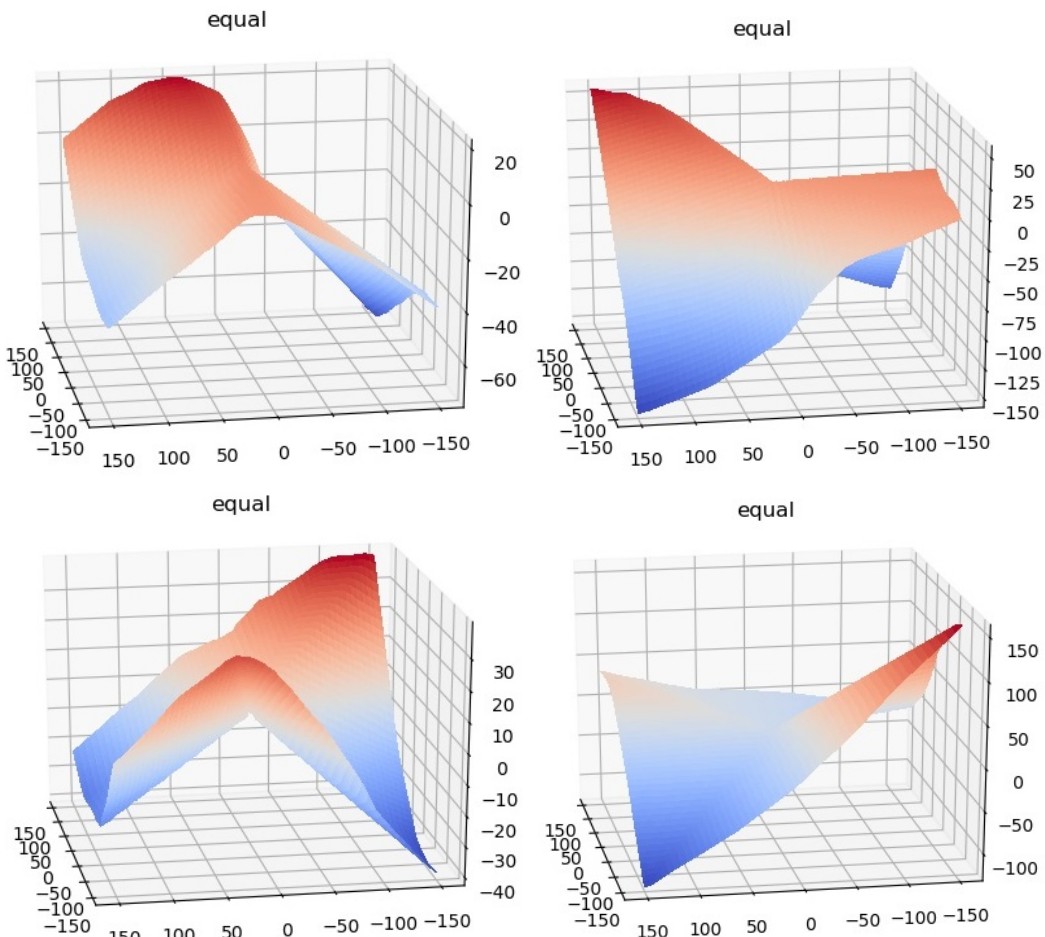

Figure 11: Plots of the 'equal' comparator function landscape for independent runs of x-coord comparison task

