# OpenReview forum: "Extrapolatable Relational Reasoning With Comparators in Low-Dimensional Manifolds"
_ICLR.cc/2021/Conference — Reject_

### Official Review · AnonReviewer1 · 2020-10-25
**Why not low-dimension?**

**Rating:** 4
**Confidence:** 4

**Review:**

The paper proposes a simple low-dimensional projection and comparison module for extrapolatable relational reasoning. The proposed module first projects object embeddings into lower-dimensional manifolds, compare and concatenate them to form a vector, before finally making a prediction. The paper shows that on the maximum-of-a-set task, the visual-object-comparison task, and the IQ test, the proposed module, when combined with deep models, outperforms existing works in extrapolation.

The paper empirically investigates how the proposed module can improve relational reasoning in ood extrapolation tasks, which is considered an important aspect for deep neural networks. The paper is well-presented, providing intuitive sketches to ease understanding.

------------------------------------

a. It looks like the major claim of the work is that low-dimension manifolds can help learning. But I wonder, for most of tasks, especially the representation used for classification, when will low-dimension manifolds hurt? Aren't the final layers of deep networks always low-dimensional? Low dimensions ease understanding, interpretation, visualization, and learning. When the true hidden representation is low-dimensional, why do we need to make it high-dimensional? I'm confused what the authors are trying to argue in this point.
b. From a technical point of view, the true lower-dimensional representation, if well-learned, will improve generalization without doubt. However, the work does not demonstrate the representation learned has this property. For example, if the authors are trying to say they learn the attribute, say size, can it be shown that learned low-dim representation is correlated with the actual size of it, independent of its position, during extrapolatory tests? And can the comparator be shown to act like a sign function. Especially in the IQ test problem? The first two tests look particularly simple, as one can even design transformations that show that. E.g., for numbers you can use identity transformation and do a subtraction and a sign function. For the black-and-white images, you can use a simple identity kernel to compute the number of foreground pixels.
c. The module design looks to be adopted directly from existing works like relational module, PrediNet, and CoPINet. Relational module and PrediNet have this object-pair and comparison idea, and the CoPINet mentions the contrast implementation. The low-dim comparator is a simple modification of the existing works.
d. While the argument is on extrapolation, I wonder what makes the module infeasible for interpolation. If the module learns well the true representation and knows what it means to compare, interpolation shouldn't be a problem either. Can it be shown that the representation is good for interpolation as well?
e. The authors, during experiments, use pre-trained models rather than training from scratch. It is possible that it's the pre-trained representation that helps the learning and extrapolation rather than the proposed low-dim projection? What if the models are not pre-trained?
f.  Finally, for Section 3.4. It looks like the low-dim representation is not well-learned. Test data representation is much clustered that the training data. Different ranges of training data are well-separated, but test data points are distributed across the test range. Also, for the comparator plot, response for training data is sure to be better than that for unseen data, but what kind of insight is there?

---

> ### Author Response · Authors · 2020-11-19
> **Reply to reviewer's comments**
>
> Thank you for your valuable comments. We address your concerns with additional experiment results below. We will also upload a revised version of our paper shortly.
>
> a. We agree that low-dimensional embedding has been researched extensively and found useful. However the focus of this paper is to apply low-dimensional embedding with a comparator to improve generalisation performance for relational reasoning tasks. To our best knowledge, the low-dimensional embeddings has not been applied for this purpose before.
>
> b. As required, we investigated if the learnt projection is correlated with the ground truth attributes for the 'Visual Object Comparison' task. We computed pearson correlation between 1-dim projected value and ground truth attributes for all three attributes. Below are the results:
> X-coord: 0.796
> size: 0.945
> color 0.977
>
> While it is possible to manually designed functions that work for the first two experiments, the point of this paper is whether neural networks can learn the necessary functions automatically, rather than requiring priors from human experts. We discussed this in the section 'Related Works' by comparing with architectures with manually designed modules such as Symoblic VQA.
>
> c. While we agree that our module is built upon architectures such as Relation Network, we would like to emphasize that this is not the main novelty of this paper. Our main novelty is to study if low-dimensional projection and comparators can be amalgamated into existing architectures for improved generalisation performance.
>
> d. The focus of this paper is extrapolation performance. Thus we didn't perform experiments on other generalisation tasks. Nonetheless during the rebuttal period we tested our model on the 'interpolation' split of PGM dataset. Our model achieved 85.3% accuracy, which is higher than all baseline models( WReN's 67.4%, MLRN's 57.8% and MXGNet's 84.6%).
>
> e. In our experiments we made a fair comparison against a baseline with the same pretrained CNN. This baseline is named 'MLRN-P' in Table 3. It can be seen that with the same pretrained front-end CNN, our model performs much better than the baseline. In addition we also performed additional experiments without the pre-trained VAE. The model achieved 21.2% extrapolation accuracy, which is still higher than all baselines.
>
> f. We are not sure if we completely understand the reviewer's point. The test data are not more clustered than training data. In fact there are more data points from training set than from test set. Figure 4.a shows that projected embeddings are in fact more clustered together into a 'line' in the 2-d space, while the projected embeddings for test data are less clustered. The insights of the comparator plot is that when the projection dimension is higher than the real dimension of compared attributes, the out-of-distribution test data will be more likely to fall outside of the comparator function space optimised during training. We are happy to discuss this further if reviewer is still in doubt.
>
> We hope we have made clear explanations to your concerns. We are happy for any further discussions or questions, so please feel free to let us know.

---

### Official Review · AnonReviewer4 · 2020-10-29
**Good idea, but the experiments don't really test the hypothesis**

**Rating:** 5
**Confidence:** 3

**Review:**

"Projecting into a lower dimensional space, so that irrelevant dimensions are projected out, will help in extrapolation" is an interesting hypothesis that should be tested. However, I was not convinced that the experiments gave convincing evidence for the hypothesis. It is standard in science to test a hypothesis by keeping as many of the extraneous variables as invariant as possible. However, in this case a different architecture (which projects onto a lower dimensional space) was used for each problem, and there was an impressive improvement. It wasn't clear to me whether it was the lower dimensionality or other architecture choices that made the difference.

Detailed question. What is g in equation (2)? I can make sense if it if the output of f is a K-dimensional vector.  But then you say "g is a function that combines....to make a prediction" so. I interpret that as producing a value. So then f gives a number, but then I can't iWork out what is the input to the MLP in equation (2); I was trying to look for a, say,  d-dimensional input, but I can't work out what d is.

The results of finding the maximum (Table 1) are dramatic. Is it really just the lower dimensional embedding? I'm guessing that you can learn the correct hypothesis, but the others can't (or there are lots of hypotheses consistent with the data).

---

> ### Author Response · Authors · 2020-11-19
> **Reply to reviewer's comments**
>
> Thank you for your valuable comments. We address your concerns below. We will also upload a revised version of our paper shortly.
>
> Q:''It wasn't clear to me whether it was the lower dimensionality or other architecture choices that made the difference.''
>
> A: In Appendix I, we performed extensive ablation study on various parts of the architecture design. It can be seen in Figure 10 that the lower dimensionality does matter in improving extrapolation performance. Other design choices are found to have effects for the PGM experiments. However for 'Visual Object Comparison' experiments we found these choices have little effects on performance.
>
> Q: ''What is g in equation (2)?''
>
> A: 'g' is a function that process the combined comparison results. In all of our experiments, 'g' is implemented as MLPs. The setup for 'Maximum of a set' problem is slightly different from the other tasks. The output of 'g' depends on the nature of the task. In equation 2, g is the 'MLP' which process the summed comparison results and output a vector e_i. This vector is then used to produce an attention value by a(e_i) used in weighted sum of the projection.
>
> Q: ''The results of finding the maximum (Table 1) are dramatic.''
>
> A: Our model has dramatically lower error because our model can more easily learn the correct hypothesis. In fact, the 'Deep Sets(Max)' model, in theory, can also learn the correct hypothesis. This model is implemented as g(max(f(x)). If the model learn 'g' and 'f' to be the identity function, then the model effectively learns the correct hypothesis. However in all our experiments, this model never converges to this hypothesis.
>
> We hope we have made clear explanations to your concerns. We are happy for any further discussions or questions, so please feel free to let us know.

---

### Official Review · AnonReviewer3 · 2020-10-29

**Rating:** 4
**Confidence:** 5

**Review:**

The paper presents two neural network design ideas: low-dimensional projection and arithmetic comparator. By integrating these two ideas into CNNs, the model can solve a set of tasks that require recognizing 1-dimensional properties of objects (such as the size, the color, etc), and making a comparison of these properties. The authors show that their method has strong out-of-distribution generalization: such as generalization to larger objects.

The presentation of the paper is relatively clear. However, I believe the authors could make the paper stronger by strengthening their experiments with more results and ablation studies. Details follow.

1. The numbers in Table 4 and Table 5 are suspicious. Can the authors verify that these two tables have exactly the same results?
2. The last sentence of the first paragraph of Page3 says "the vector distance p(o_i) - p(o_j) or absolute distance improves the generalization performance". Are there any ablation studies for those?
3. The sec 3.1 "maximum of a set" experiment is toy-ish: in some sense, only the proposed model takes comparison results as the model input, while all other models take numbers. The authors should consider other baselines that also take comparison results as their inputs.
4. I suggest the authors discuss more the results in Table 2: what are the failure cases of the model? Also in Table 2, the author should make ablation studies to discuss the importance of 1) low-dim embeddings and 2) comparison operations. For example, what will the baseline perform if it also projects the input image into a very low-dim space?
5. Table 3. I suggest the authors supply the performance on standard train-test splits for RPM as well. This will give us an idea of how the model performs on "i.i.d." test splits. Also, what are the failure cases for Table 3?
6. I have some concerns about the applicability of this proposed model. Learning meaningful low-dimensional embeddings can be hard. For example, how to extract the number of people in an image? Or in an even simpler task, how to count the number of vertices of an arbitrary polygon in an image? The authors should discuss the limitation of their method.

---

> ### Author Response · Authors · 2020-11-19
> **Reply to reviewer's comments**
>
> Thank you for your valuable comments. We address your concerns with additional experiment results below. We will also upload a revised version of our paper shortly.
>
> 1. Thank you for pointing out this error. The correct accuracies for Table 5 should be 23.5, 25.1 and 25.9. This will be fixed in our revised version of the paper.
>
> 2. We performed additional ablation study of the comparator input for the visual object comparison task. Specifically we measured the extrapolation test accuracies for: 1. vector distance 2.absolute (l1) distance and 3 learnt distane with a MLP. Below are the results:
> vector dist accuracy:0.9771 +- 0.0281
> l1 dist accuracy:0.692std +- 0.149
> MLP dist accuracy:0.912 +-0.0652
>
> 3. Regarding baselines that also take comparison results as input, we would like to point out that in the baseline 'Set Transformer', elements in the set are compared with the attention mechanism of Transformer architecture. In addition to this, we experimented with a hierarchical Relation Network architecture that takes comparison results as input. This model achieves 16.2 +- 2.45 M.S.E. We will include the detailed architecture in our revised paper.
>
> 4. Regarding failure cases of visual object comparison, we performed manual inspection of test results, and found that the incorrect predictions happen more frequently when test values are further from the training range. This is an expected results as the further the test data, the further are the embeddings from trained function space.  Ablation study of low-dimension embedding is already included in Appendix I. We performed additional ablation study of the comparison functions with different input distance metric. The results are listed in our answer to the 2nd point of reviewer.
>
> 5. We additionally trained our model on the 'neutral' split, which is the only i.i.d split in PGM datasets. Our model achieves 90.1% accuracy. We can only perform limited failure cases analysis as PGM dataset does not provide ground truth labels about object attributes. We can only identify the trend that the error rates increase with more parallel relations in a task (which means it is more difficult).
>
> 6. We performed experiments in 3 different relational reasoning tasks in hope of illustrating the applicability of our module across a range of relational reasoning tasks. There are definitely many other tasks we cannot address, but our module can be simply used in any architecture that performs relational reasoning. The simplest example will be swapping the pairwise MLP in Relation networks with our module.
>
> We hope we have made clear explanations to your concerns. We are happy for any further discussions or questions, so please feel free to let us know.

---

### Official Review · AnonReviewer2 · 2020-11-02
**initial review**

**Rating:** 5
**Confidence:** 4

**Review:**

Review:
This paper addresses an inductive bias for relational reasoning tasks to improve generalization performance on out-of-distribution scenarios (so called extrapolation), which the value ranges (continuous variables) or value itself (discrete variables) of the training/test dataset do not overlap. The proposed idea is to project high-dimensional representations onto low-dimensional space and compare those of the targets of relational reasoning. To validate effectiveness of the idea, the authors provide the improved results on three problems: (1) finding the maximum of a set of real numbers, (2) comparing the attributes of visual objects with dSprites dataset, and (3) visual reasoning with Raven progressive matrices (RPM) based on the PGM dataset. The experimental results of each  show better performance than other baseline methods. This research deals with one of important topics in AI/ML research. Out-of-distribution generalization is potentially extended to world models, common sense reasoning, learning and reasoning with small number of instances, and abstraction capability of human intelligence. As Section of related work in this paper also mentions these points, it is understandable to figure out the position of this work.
On the other hand, I think that the idea of projection and comparison is similar to the approaches of zero-shot learning and deep metric learning. So, the novelty of this idea is not so big and it would be better to explain them as related work. Considering the experimental results in Section 3.4, it is not sure whether the proposed method still shows similar efficacy regardless to the degree of difference between the training datasets and the out-of-distribution ones.
I recommend as ‘Marginally below acceptance threshold’

Pros:
- In Section 3.4, it is interesting that the analyses of the projection space and low-dimensional embedding and why the dimension should be small. I like it.
- The authors propose a generalized improvement strategy for relational reasoning tasks on the out-of-distribution scenarios, and they show that the key idea effectively improves the performance.

Concerns:
- The proposed idea is not so concrete to apply as practical solutions. I think it seems close to a guideline. If there is someone to use this approach, it needs lots of effort and time for the actual problems.
- It seems that the distribution difference between training datasets and the out-of-distribution ones influences explicitly on their performance. Does the main claim still valid in that situation? What is the relationship between the difference and the validness?
- How does the dataset for visual object comparison split? Overall, there are not much explanation on details of the configurations of comparative methods. It seems difficult to reproduce the results.
- What is the baseline method of Figure 5 (b)?
- Overall presentation is not so clear. That makes difficult to follow.

Minors:
- I think ‘Less defined’ is not so good expression.
- Figure 4 (b) has no information on each axis.
- The right end of Figure 5 (a) is covered by Figure 5 (b).
- Figure 5 (b) shows poor readability.

---

> ### Author Response · Authors · 2020-11-19
> **Reply to reviewer's comments**
>
> Thank you for your valuable comments. We address your concerns below. We will also upload a revised version of our paper with clearer presentation shortly.
>
> Q: ''it is not sure whether the proposed method still shows similar efficacy regardless to the degree of difference between the training datasets and the out-of-distribution ones''
>
> A: In fact Figure 5 (b) illustrates the effectivness of our model for different degree of diffrence between training and test set. The 'Beta', the truncation ratio (See section 2.5), measures the difference between training and test set. It is true that as the test data distribution are further from the training distribution, all models' performance degrade. However our model's performance deteriorate much less than baselines.
>
> Q: ''The proposed idea is not so concrete to apply as practical solutions. I think it seems close to a guideline.''
>
> A: We agree with reviewer that the main point of this paper is not to develop a concrete architecture that achieves SOTA results on certain datasets. Rather we focus this paper in promoting the effectiveness of low-dimensional comparison in relational reasoning tasks, and to propose theoretical tools that can more properly measure out-of-distribution generalisation performances.
>
> Q: "It seems that the distribution difference between training datasets and the out-of-distribution ones influences explicitly on their performance. Does the main claim still valid in that situation? What is the relationship between the difference and the validness?"
>
> A: As answered above, Figure 5(b) shows that all models' performance degrade as the test data distribution are further from the training distribution. However our model's performance deteriorate much less than baselines. Thus our claim that 'low-dimension comparison improves extrapolation performance' is still valid.
>
> Q: "How does the dataset for visual object comparison split? How does the dataset for visual object comparison split? Overall, there are not much explanation on details of the configurations of comparative methods. It seems difficult to reproduce the results."
>
> A: This is discussed in section 3.2. We apologize if the description isn't clear. We will include a more detailed description in our revised version. Detailed architecture configurations are described in Appendix A,C and E. Regarding reproducibility, we will release our code upon acceptance to this or other conferences.
>
> Q: "What is the baseline method of Figure 5 (b)?"
>
> A: The baseline model is the same as used for experiments in section 3.2, which is a simple MLP model.
>
> We hope we have made clear explanations to your concerns. We are happy for any further discussions or questions, so please feel free to let us know.

---

### Official Review · AnonReviewer5 · 2020-11-05
**Good results and analysis but description requires work**

**Rating:** 6
**Confidence:** 4

**Review:**

This paper showcases three studies on relational reasoning where objects need to be compared on a certain attribute (like size). The experiments show large improvements in generalization performance over previous work. The authors attempt to formulate a general method from their experiments.

My biggest criticism is that the method needs to be described more clearly. There are lots of variations and add-ons from one experiment to the next. When you introduce notation, it is not clear what it maps to in your architecture descriptions (e.g. the choice of g and c). Is it fair to say that the main insight of the paper is the following: it is possible and desirable (in terms of generalization) to learn comparators between objects by feeding the difference between low-dimensional (or specifically one-dimensional) projections of the object representations, taken pairwise, into small MLPs?

The hyperparameters/architectural choices need to be summarized better. Could you perhaps make a table specifying the following for each experiment (I’ll refer to maximum of a set, dSprites, and RPM as experiments 1, 2, and 3 respectively):
- the value of K (this is clear for experiment 1 but not others). Appendix E seems to suggest you use K = 512 for experiment 3. This is much larger than what I would expect.
- the size of the low-dimensional manifold for each projection p. Is this always 1? Figure 10 seems to show 1 works best, which is a nice result.
- the comparison summarizer g
- the exact form of the comparator function c. Are you feeding in p(o_i) - p(o_j) into an MLP or |p(o_i) - p(o_j)| or something else altogether? This isn't clear especially for experiment 1.
- how many objects are you ultimately comparing/learning a relation over. If I understand correctly, this is N, 2, and 8 for the three experiments respectively. Please do specify N.
- what is the final output. This is clear for experiment 2 and 3. For experiment 1, I assume the output is a categorical probability vector with N dimensions? Or is it in fact the value of the maximum element?
- what type of loss you are using. You defined a fairly non-intuitive loss L(d(a_t(o_i), a_t(o_j)), f(o_i, o_j)) in section 2.1. It is unclear where you’re using this.

The strengths of the paper include:
- Solid results showing better strictly better generalization
- Good attempts to analyze the method

Minor: I do also take exception when you refer to your proposed comparator as “an inductive bias module.” (I realize there is no consensus on the definition of the term “inductive bias.” If we do define it–as something which changes the preferred solutions of a learning algorithm–then most deep learning architecture choices are inductive biases. So it is rather vacuous to refer to your architectures as inductively biased.) Moreover, an extra input (e.g. p(o_i) - p(o_j) which is fed to the comparator in your case) cannot be termed as an inductive bias, because it changes the solution space completely (consider learning the XOR function with a single linear layer when the difference is provided as an additional input versus when it is not).

Lastly, the inspiration from neuroscience is quite loose. The fact that human brains have high-level neurons which disentangle object position and size does not lead to your method directly; you could argue that all of deep learning attempts to learn higher-level features progressively with depth in the network. I would suggest revising the paper to focus on the insights, along with a clear description of the method, rather than the loose inspiration.

Questions and request for clarification:
1) Could you please provide error bars for the RPM experiment?
2) On your observations (section 3.4) answering why comparators on low-dimensional manifolds improve generalization:
- I would suggest moving Figure 10 from the appendix to this section to answer why low-dimensional projects are important to learn good comparators.
- I see that observation 1 holds for 2D projection spaces, but could you illustrate the linearity of the projection in 3+ dimensions? Alternatively, please change the phrasing to refer to 2D specifically to avoid overclaiming.
- Under observation 3, your claim that the 2D comparator is “random [...] outside of the training points’ sub-manifold” in Figure 4b: in what sense do you mean that? Could you add similar plots over independent runs so the “randomness” is evident?
3) I do appreciate your efforts to show systematic generalization. But your o.o.d algorithmic alignment metric is sloppy and arbitrarily defined. For instance:
- What is V in the definition? I don’t see it defined or used anywhere in section 2.5.
- What are the requirements on \Tau, the truncation function? It seems to be constrained by \beta and \textbf{u} but this isn’t fully specified. I suppose you want to ensure the truncation function is not vacuous (e.g. the identity function)?
- Are all dimensions in \textbf{u} truncated by the same \beta parameter? Why choose a subset of dimensions for truncation at all (rather than truncate all dimensions)? This needs justification.
- How would the truncation work when the training data are images?
- Given than D_s is only a truncation of D, it seems fair to say that the sample {x^s_i, y^s_i} has non-zero likelihood of coming from D instead of D_s. So strictly speaking, your definition of (M, \epsilon, …)-learnability is not measuring o.o.d performance; rather it seems to be measuring performance on a “superset” of the training sample. Is this a correct reading?
4) Your algorithmic alignment results (section 3.5) show that your comparator generalizes favorably over the baseline for various values of \beta and the sample size. But what dataset are the scores based on? What is the truncation function? What is the baseline?
5) Notation: could you please avoid overloading the use of "a" as the ground-truth mapping function (from an object's representation to its t'th attribute) as well as the attentional weights for pooling? Ideally, avoid defining the former unless you're actually using it.

In conclusion, I do consider this work substantial and important. I would be open to revising my score if the presentation is sharpened.

---

> ### Author Response · Authors · 2020-11-19
> **Response to Reviewer's comments**
>
> Thank you for your insightful and valuable comments. We address your concerns with additional experiment results below. We also worked on improving the presentation. We will also upload a revised version of our paper shortly.
>
> Q: "Is it fair to say that the main insight of the paper is the following: it is possible and desirable (in terms of generalisation) to learn comparators between objects by feeding the difference between low-dimensional (or specifically one-dimensional) projections of the object representations, taken pairwise, into small MLPs?"
>
> A: This is a very accurate statement. Our main goal is to illustrate the effectiveness of low-dimensional comparators for a range of relational reasoning task, and develop a measurement tool based on algorithmic alignment theory to measure model's alignment for generalisation performance.
>
> Q: "Could you perhaps make a table specifying the following for each experiment":
>
> A: We described the hyper-parameters in Appendix A,C and E. But indeed tables might be a clearer way to present these. Below are the required hyper-params in crude format. We will also include these tables in our revised paper.
>
> Experiment 1:
> K=1, p-dim=1, comparator= MLP(p(o_i) - p(o_j)), num_objects=20, output=maximum_val (real valued), Loss function=Mean Squared Error
>
> Experiment 2:
> K=1, p-dim=1, comparator= MLP(p(o_i) - p(o_j)), num_objects=2, output=categorical, Loss function=Cross-Entropy
>
> Experiment 3:
> K=512, p-dim=1, comparator= MLP(p(o_i) - p(o_j)), num_objects=8 Context + 8 Ans, output=categorical, Loss function=Cross-Entropy
>
> Note that for experiment 3, K is large because we are comparing diagrams that contains multiple objects with many different attributes. Thus a large K is needed for all attributes of these objects.
>
> Q: "Could you please provide error bars for the RPM experiment?"
>
> A: We only had time for 5 runs. The result is: 25.2 +- 0.59
>
> Q: "On your observations (section 3.4) answering why comparators on low-dimensional manifolds improve generalization":
>
> A:
> 1. We originally have figure 10 in the main text, but moved it to appendix due to space limit. We will move it back as the revised version is allowed 1 extra page.
> 2. It is difficult to visualise the function space of 3+ dims. We will emphasize in our revised version that the visualization is only for 2-d case. However we developed a quantitative measure based on hausdorff distance to measure divergence between training and test embeddings for higher dimensional cases.
> 3.  We have generated additional plots of the function landscape for 5 independent runs. We will include the plots in our revised paper as we are not allowed to include figures in this comment.
>
> Q: "o.o.d algorithmic alignment metric is sloppy and arbitrarily defined":
>
> A:
> 1. We are sorry that we didn't define V clearly. 'V' is the value range of the whole dataset distribution, as used in section 2.2.
> 2. The truncation function 'T' can be any functions that truncate the data distribution given a truncation ratio '\beta' and truncation dimensions 'u'. The truncation can be performend in the data distribution (e.g. select one dimension of the data space and truncate it to be in range (0,beta). In the 'Visual Object Comparison' task, we truncate the latent data distribution instead. The latent data distribution of dSprites dataset(according to the beta-VAE and dSprites dataset paper) has six latent dimensions which are: (color, shape, size, rotation, x-coord, y-coord). For example, we can select the x-coord dimension (range [1,32]), and truncate the distribution with
> beta=0.6 so the samples from the truncated distrubtion can only have x-coord in range[1,32*0.6].
> 3. "rather it seems to be measuring performance on a “superset” of the training sample. Is this a correct reading?"": This is correct. The definition can also be easily modified for o.o.d case. One just need to change D in equation 3 to "D \ D_s". However we feel the current definition is more realistic as real-world test data are not completely out-of-range, but just have some outliers that are o.o.d.
>
> Q: "But what dataset are the scores based on? What is the truncation function? What is the baseline?"
>
> A: We did say that the analysis is on 'size comparison task' in section 3.5. So the dataset is the same as in Visual Object Comparison task. The truncation function is described in pt.2 of the above answer, which truncates the latent data distribution. The baseline is also the same baseline used in 'Visual Object Comparison' task.
>
> We hope we have made clear explanations to your concerns. We are happy for any further discussions or questions, so please feel free to let us know.

---

### Author Response · Authors · 2020-11-20
**Revised Version**

Thanks reviewers for their detailed comments. We have uploaded our revised paper to address many concerns raised. The major changes are:

1. Clearer explanation in section 2.5 about our proposed algorithmic alignment metric. Added Appendix section K to describe the exact truncation function used in definition 2.1.
2. Clearer explanation in section 3.5 with more details on dataset and baseline used. More explanation about Figure 5.b.
3. Added Section L in Appendix to discuss correlations between projected embedding and ground truth attributes of objects.
4. Added section M in Appendix to describe experiment results on 'neural' and 'interpolation' split of PGM dataset.
5. Fixed errors in Table 8 (originally Table 5).
6. Added results of an additional baseline (2-Level Relation Network) in Appendix N. for Maximum of a set task that explicitly takes comparison results as input.
7. Added additional ablation study on the type of comparator functions (Table 9, Appendix I).
8. Added more plots of the comparator function landscape for independent runs in Appendix O.
9. Moved the ablation study of projector dimensions from Appendix I to section 3.4 of the main paper.
10. Improved connection from neuroscience inspirations to our proposed model, and added more insights in the Introduction section.
11. Other improvements in presentation such as improving explanations and fixing symbol overloading.

---

### Decision · Program_Chairs · 2021-01-07
**Final Decision**

**Decision:**

Reject

**Comment:**

This paper is attempting to improve the OOD generalization performance of neural networks on relational reasoning tasks. This is an important failure point of general neural network architectures and important research topic. The results of the paper shows impressive improvements on a set of subject.

* The paper is improved during the rebuttal, however, I do agree with the R5 and the paper is still lacking a lot in terms o clarity. The writing of this paper still requires some work.

* As R2 also has written, the proposed idea is not so concrete to apply as practical solutions, and the presentation of the paper still requires some more work.

* R3 pointed out some inaccuracies and it seems like authors have added some ablations in the direction that R3 has suggested.

I am suggesting to reject this paper given that the majority of the reviewers are also leaning towards rejection as well. I would recommend the authors to improve the clarity of the paper, do more ablations for their models and resubmit to a different conference.